# Proliferation and Apoptosis Pathways and Factors in Oral Squamous Cell Carcinoma

**DOI:** 10.3390/ijms23031562

**Published:** 2022-01-29

**Authors:** Steven He, Rajdeep Chakraborty, Shoba Ranganathan

**Affiliations:** Applied Biosciences, Faculty of Science and Engineering, Macquarie University, Sydney, NSW 2109, Australia; steven.he@hdr.mq.edu.au (S.H.); rajdeep.chakraborty@hdr.mq.edu.au (R.C.)

**Keywords:** proliferation, apoptosis, oral cancer, apoptotic factors, proliferation factors, tumor suppressor

## Abstract

Oral cancer is the most common form of head and neck squamous cell carcinoma (HNSCC) and most frequently presents as oral squamous cell carcinoma (OSCC), which is associated with an alarmingly high mortality rate. Internationally, a plethora of research to further our understanding of the molecular pathways related to oral cancer is performed. This research is of value for early diagnosis, prognosis, and the investigation of new drugs that can ameliorate the harmful effects of oral cancer and provide optimal patient outcomes with minimal long-term complications. Two pathways on which the progression of OSCC depends on are those of proliferation and apoptosis, which overlap at many junctions. Herein, we aim to review these pathways and factors related to OSCC progression. Publicly available search engines, PubMed and Google Scholar, were used with the following keywords to identify relevant literature: oral cancer, proliferation, proliferation factors, genes, mutations, and tumor suppressor. We anticipate that the use of information provided through this review will further progress translational cancer research work in the field of oral cancer.

## 1. Introduction

Globally, oral cancer is the most common head and neck malignancy with an estimated 34,864 new global cases in 2018 rising to 377,713 in 2020, more than doubling the 185,976 number cases observed in 1990 and demonstrating its rapidly increasing prevalence [1,2,3]. Melanesia and Southcentral Asia exhibit the highest incidence of oral cancer—accounting for approximately 52% of global cases—followed by Eastern Europe and then Australia/New Zealand [3].

Oral squamous cell carcinoma (OSCC) is the most common head and neck squamous cell carcinoma, accounting for 90% of all oral cancers [4], with an estimated 5-year overall survival rate of only approximately 50% [2,5]. Various treatment options exist, including surgery, radiotherapy, chemotherapy, or a combination thereof, often accompanied with post-operative follow-up and monitoring [6,7]. Treatment of OSCC during its early stages shows the most favourable prognosis, with an estimated 92% 3-year survival rate [8], highlighting the vital importance of early detection.

Many associated aetiological factors of oral cancer have been well established, with risk predominantly increasing with age, alcohol and tobacco use [1,9,10]. While the International Agency for Research on Cancer (IARC) classifies tobacco as a Group I carcinogenic substance in the oral cavity [11], the mechanism for pre-disposing risk with alcohol is less clear; it has been speculated that ethanol is metabolised by oral microflora and transformed into acetaldehyde, representing a carcinogenic substance [12]. As evidenced by the previously mentioned prevalence, oral cancer risk is also exacerbated in Melanesia and Southcentral Asia due to the traditional chewing of areca nuts (also known as betel nut), which are used in the preparation of betel quid, a substance also identified as a Group I carcinogen due to the high concentration of alkaloids present in the nut [13]. Human papilloma virus (HPV) infection has also been classically recognised as a risk factor due to the histological similarity of the oral and vaginal mucosa [14], high prevalence of HPV in OSCC [15], and the ability of HPV to immortalise human keratinocytes in vivo [16]. However, discrepancies regarding this classification exist and remains controversial. While there is compelling evidence for HPV as a risk factor in oropharyngeal cancer [17], HPV DNA presence in potentially oral malignant lesions has been reported to range from 0 to 85% [15], with additional reports of HPV-positive cancer rates as low as 6–13% [18,19]. It has been alternatively postulated that HPV infection may be opportunistic and not necessarily a cause of carcinogenesis [20].

OSCC often develops from dysplastic precursor lesions in the epithelium, which can be characterised by loss of apical-basal polarity in epithelial cells, increased nuclear-cytoplasmic ratio, irregular epithelial stratification, loss of intercellular adherence, nuclear pleomorphism—such as enlarged nuclei or nuclear hyperchromatism—and abnormal keratinisation [21,22]. The most common form of dysplastic lesion is oral leukoplakia (OLK) and its more aggressive clinical variant, proliferative verrucous leukoplakia [4]. OLK is only loosely defined by the World Health Organisation (WHO) as, “A white patch or plaque that cannot be characterised clinically or pathologically as any other disease”, highlighting the need for further genetic and molecular research in this area, given that these lesions are likely pre-malignant and associated with a 40.8-fold increased risk of development into oral squamous cell carcinoma [23,24].

## 2. Molecular Hallmarks of Cancer

Six hallmarks of cancer have been identified to provide a broad framework for characterising the complex multi-step development of this disease [25]. We provide a brief overview here for the three hallmarks: enabling replicative immortality, inducing angiogenesis, and activating invasion and metastasis, with more detailed review of the proliferation and apoptosis signalling pathways in subsequent sections. The subsequent sections address remaining hallmarks; sustaining proliferative signalling; and evasion of growth suppressors and resisting cell death due to their heavier implication in oral cancer biology.

### 2.1. Enabling Replicative Immortality in Oral Cancer

In many normal cell lineages, there is a limited number of growth and division cycles that a cell can undergo before reaching cellular senescence—where the cell is viable but non-proliferative—or crisis, where cell death occurs [25]. This is dictated by telomeres, which are long stretches of DNA tandem repeats that cap the ends of chromosomes and shorten with successive cell division cycles. Telomerase is a specialised DNA polymerase that elongates telomeric DNA, allowing further cellular division by synthesising additional telomere repeat sequences and, while absent in most non-immortalised cells, is functionally expressed in over 90% of immortalised cells, including human cancer cells [25]. In an immunohistochemical study of patient OSCC and oral epithelial dysplasia samples, 81.48% and 77.06% of cells within tissues were observed to express telomerase, with only 62.91% activation observed in normal oral mucosa controls [26].

### 2.2. Inducing Angiogenesis in Oral Cancer

In tumour tissue, genetic changes often result in the constitutive activation of angiogenesis to provide oxygenation, nutrients, and the removal of metabolic waste to enable growth and proliferation. These new blood vessels often develop through dysregulation of a small subset of cytokines, specifically through upregulation of vascular endothelial growth factor (VEGF)-A and decreased expression of its inhibitor, thrombospondin-1 (TSP-1) [25]. Factors associated with angiogenesis, such as CD44, and blood microvessel density have correspondingly been reported to increase in oral cancer compared to normal epithelium [27].

### 2.3. Activating Invasion and Metastasis in Oral Cancer

As epithelial cancers progress towards more severe stages, they develop morphological and molecular changes that allow them to further invade local and distal tissues. One of the best characterised changes is the loss of E-cadherin, which is a key cell-to-cell adhesion molecule that assists in sheet assembly of epithelial cells and helps maintain cellular quiescence [25,28]. Indeed, meta-analyses have reported poorer overall prognosis in OSCC patients with reduced E-cadherin expression compared to those with normal or elevated expression levels [29]. As cancer cells further develop, dysregulation of additional transcription factors may result in epithelial-mesenchymal transition (EMT), resulting in resistance to apoptosis, expression of matrix-degrading enzymes (such as metalloproteinases), and increased motility [25,30].

## 3. Oral Cancer and Cellular Proliferation

In the adult human body, the majority of cells exist in non-proliferative states, being either terminally differentiated or existing in a quiescent state (classified G_0_), with the exception of a small pool of stem-transit amplifying cells which exist in self-renewing tissues such as the epithelia [31]. Cellular proliferation itself is a complex and tightly regulated process implicating many different proteins and multiple pathways that are often able to influence each other, with dysregulation in these often being implicated in cancer development.

The following sections review the disruptions in proliferative signalling observed in oral cancers. A schematic summary outlining these signalling pathways under normal conditions can be viewed in Figure 1, with further details of key proteins and their functions available in Table 1.

### 3.1. Mitogenic Activation and Induction of Proliferation Signaling in OSCC

In healthy cells, there is careful control over the production and release of growth signals such as epidermal growth factor (EGF) and transforming growth factor α (TGFα), allowing for cell number homeostasis and maintenance of normal tissue architecture [25]. As one of the prime hallmarks of cancer, tumour cells are able to deregulate these controls to sustain their proliferative signalling. This dysregulation may be paracrine or autocrine, by which cancer cells can stimulate normal cells within the tumour microenvironment to release growth factor ligands or they may self-produce them, usually accompanied by elevated cognate receptor expression to allow for ligand hyper-responsiveness [25,66]. Genetic profiling of oral cancer samples has demonstrated this hyper-responsiveness through recurrent focal amplifications in *EGFR* and *ERBB2* genes, respectively, encoding for receptor tyrosine kinases (RTKs) epidermal growth factor receptor (EGFR) and erythroblastic oncogene B 2 (ERBB2, alternatively known as human epidermal growth factor receptor 2; HER2) which respond to mitogenic ligands [67]. Increased transcription of *TGFA*, encoding for the cognate ligand TGFα, has also been reported in vitro in oral cancers [68].

### 3.2. Ras-Raf-MEK-ERK/MAPK Pathway

The dimerization of RTKs results in the activation of Ras by Son of sevenless 1 (SOS1) via exchange of guanosine diphosphate (GDP) for guanosine triphosphate (GTP) [33]. Ras is a key mediator of proliferative signalling not only in the Ras-MAPK axis but also through possible direct activations of phosphoinositide 3-kinase (PI3K) in the alternate PI3K-Akt signalling axis [69]. Being a key regulator, Ras is genetically deregulated in over 20% of oral cancers, either through genetic mutation and/or amplification [70]. By contrast, Raf and MEK, both downstream effectors of Ras, show drastically lower mutability in oral cancer. By conducting exon sequencing of a small cohort of OSCC samples, a mutation rate of 2.4% has been reported for Raf [71], with little or no data reported on MEK mutation in oral cancer to date. In contrast to Raf and MEK, extracellular signal-regulated kinase (ERK, also known as mitogen-activated protein kinase; MAPK) has been reported to have over 100 downstream cytoplasmic and nuclear targets, including transcription factors that drive the expression of D-type cyclins which initiate the cell cycle [36,72]. Whilst overexpression of ERK has been reported in OSCC [5,73], the reported incidence of this is low compared to mutations in the *RAS* gene [74]. Interestingly, in silico pathway analysis found that both *ERK/MAPK* and *RAS* (along with *AKT* and *mTOR*) expression levels were substantially lower in pre-cancerous OLK compared to OSCC, resulting in speculation that mutation in these particular genes results in acquisition of the cancer phenotype [5]. Further compelling evidence for dysregulation of this pathway being involved in tumorigenesis can be observed in the development of OSCC in transgenic mouse models. Two models have been described where overexpression of *KRAS*, a member of the Ras family, coding for the K-Ras protein that is part of the RAS/MAPK pathway (shown in Figure 1) in the oral epithelium resulted in the growth of premalignant oral papillomas [75] or dysplasia and squamous cell carcinoma [76].

### 3.3. PI3K-AKT-mTOR Pathway

Phosphoinositide 3-kinase (PI3K) is another major downstream effector of the ErbB family of RTKs and phosphorylates phosphatidylinositol-4,5-bisphosphate (PIP2) into the second (or secondary) messenger phosphatidylinositol-3,4,5-triphosphate (PIP3) [36]. Multiple lines of evidence heavily implicate PI3K in the carcinogenesis of the oral epithelium [77,78,79]. In an in vitro next-generation sequencing study, mutations in *PIK3CA*—encoding for the p110 catalytic subunit of PI3K—were observed in 7% of 170 oral pre-cancer patient samples, with additional independent sequencing studies of 279 oral cancer samples consistently identifying *PIK3CA* among the top mutated genes [67]. Exacerbating this dysregulation, *PTEN*, which encodes for phosphatase and tensin homolog (PTEN), an inhibitor of PIP3 formation via dephosphorylation into PIP2, was similarly identified as one of the top mutated genes with somatic mutations commonly resulting in downregulation of PTEN protein expressions [67]. The importance of PTEN has also been demonstrated in vivo, where the inducible loss of PTEN and TGFBR1 (encoding for type I transforming growth factor β receptor) in transgenic mice resulted in epithelial hyperproliferation and visible carcinoma formation 10 weeks after induction with tamoxifen [80]. PTEN and PI3K exert strong influence over the downstream AKT/mTOR pathway which, in addition to its canonical function in proliferative signalling [36], also has effects on apoptosis, migration, and metabolism [51,81].

### 3.4. The Cell Cycle and Tumour Suppressor Genes

The cell cycle is the series of cellular events that results in the production of two genetically identical daughter cells from a parent cell and is largely driven by specialised groups of proteins known as cyclins and cyclin-dependent kinases (CDKs) [82]. A schematic summary of the multiple phases of the cell cycle can be viewed in Figure 2.

Retinoblastoma protein (RB), a member of the pocket protein family, and p53 are both key tumour suppressors that regulate the cell cycle. RB is a major G_1_ checkpoint protein capable of inhibiting E-type cyclin production and the E2F family of transcription factors which regulate the expression of proteins necessary for DNA replication [40]. The additional tumour suppressor, p53, signals many downstream effectors, including cyclin-dependent kinase inhibitors (CKIs) p21, p27, and p57, which are able arrest the cell at G_1_/S to allow DNA damage to be repaired or, otherwise, if the damage is irreparable, initiate apoptosis of the cell [40,83]. The *TP53* and *RB1* genes, which encode p53 and RB respectively, are commonly observed in vitro to be structurally altered in patient oral cancer samples [67], with an estimated 29% of pre-cancerous oral lesions demonstrating mutations to *TP53* [84]. The importance of these two tumour suppressor genes has been highlighted in vivo through transgenic p53-deficient mice expressing human cyclin D1, an inhibitor of RB, which develop invasive oral-oesophageal squamous cell carcinoma by approximately 6 months of age [85]. The formation of well-differentiated squamous cell carcinoma in the oral cavity, as well as skin, has also been observed more recently in transgenic mice containing only deletions of *p53*, although these carcinomas developed later with an average latency of 15–16 months [86]. Structural alterations such as deletions and fusion events are also frequently observed in *CDKN2A*, a gene encoding for p14 and p16 which prevent both RB inactivation and p53 degradation [67,87]. An estimated 15% of pre-cancerous lesions show some form of *CDKN2A* mutation [84], with further observations of transcriptional silencing due to high frequency hypermethylation in promoter regions strongly implicating *CDKN2A* in oral carcinogenesis [88]. The loss of these tumour suppressors is one of the classical hallmarks of cancer, resulting in unrestrained cell cycling and dysregulated cell growth [25].

Although the complex cellular proliferation network (Figure 1 and Figure 2) is tightly regulated, OSCC mutations can occur at many junctions within this process. These mutations are at the level of mitogenic signalling, during downstream signal transduction through multiple pathway axes or at the cell cycle level predominantly by altering tumour suppressor activity. A greater understanding of the most common mutations will undoubtedly benefit the development of new OSCC treatment options.

## 4. Oral Cancer and Apoptosis

Apoptosis is the process of programmed cell death of which three main pathways exist: the intrinsic, the extrinsic, and the granzyme B pathways. All three pathways ultimately result in the activation of caspase proteins that trigger a proteolytic cascade to dismantle and remove the dying cell [89]. This process is a vital component of healthy cell turnover and tissue homeostasis and acts as one of the critical barriers guarding against cancer development, with resistance to cell death constituting one of the hallmarks of cancer [25]. A schematic summary outlining the three primary apoptotic pathways under normal conditions can be viewed in Figure 3 with further information on key proteins available in Table 2.

**Table 2 ijms-23-01562-t002:** Summary of apoptotic proteins and their role in oral cancer.

Protein	Role in Oral Cancer	Reference
Granzyme B	Cytotoxic T lymphocyte mediated tumour cell apoptosis	Zhu et al. [90]
Perforin	Takes part in NK cell mediated oral cancer cell destruction	Hadler-Olsen et al. [91]
FasL	Mediator of immune privilege in OSCC	Fang et al. [92]
Fas (CD95)	Considered a prognostic marker of OSCC	Peter et al. [93]
TNF-α	Promotes oral cancer growth and pain	Salvo et al. [94]
TNFR1 (CD120a)	Related to oral cancer pain and inflammation	Scheff et al. [95]
FADD	Prognostic implication in OSCC	Gonzales-Moles et al. [96]
Apaf1	Helps in the formation of apoptosome	Dwivedi et al. [97]
Bcl-2	Altered expression results in an increase in malignant transformation potential	Juneja et al. [98]
Bcl-2a1 (Bfl-1) and Cytochrome C	Promotes apoptosis in OSCC	Zheng et al. [99]
Mcl-1	Overexpression of anti-apoptotic factor Mcl-1 leads to progression of OSCC	Sulkshane et al. [100]
Bcl-xL	Causes resistance to chemotherapeutic drugs	Alam et al. [101]
Bcl-w	Cooperates with oncogene activation in the development and progression of OSCC	Hartman et al. [102]
BH3	Anti-anti-apoptotic functions in OSCC	Carter et al. [103]

### 4.1. Extrinsic Apoptotic Signalling Receptors and Ligands

Dysregulation of the extrinsic pathway of apoptosis appears to be one of the primary mechanisms by which oral cancers are able to resist cell death. The expression of the Fas receptor is suppressed in OSCC, with two separate studies reporting detectable Fas in less than 5% of experimental OSCC samples in vitro [110,111]. Of interest, these studies also identified increased OSCC expression of Fas lignad (FasL), corroborating findings that OSCC samples secreted FasL positive membranous vesicles that were capable of inducing apoptosis in activated T lymphocytes [112]. The resultant effect is two-fold with OSCC cells being able to avoid cell death not only by downregulating Fas expression but also by upregulating and secreting FasL positive vesicles to induce apoptosis of T lymphocytes, which normally act as anti-cancer agents. It has also been previously reported that Fas receptor gene polymorphisms are correlated with increased malignant potential of oral submucous fibrosis, another form of oral lesion [113].

In addition to Fas receptor mutations, genome characterisation has observed frequent inactivation of the gene encoding TNF receptor associated factor 3 (TRAF3) [67], which interacts and mediates signal transduction from members of the TNF receptor family—another key receptor family involved in extrinsic apoptotic signalling. In a similar fashion, polymorphisms in the cognate ligands of the TNF receptors tumour necrosis factor alpha (TNFα) and beta (TNFβ) have also been reported to increase the risk of oral cancer in European populations [114].

Downstream of these death receptors, amplification of the Fas-associated protein with death domain (FADD) gene in 38% of head and neck squamous cell carcinoma has also been reported [67]. While upregulation of FADD seems counterintuitive to apoptotic resistance, selection of this gene may instead relate to its other pleiotropic non-classical functions in inflammation, differentiation, and cell growth, with both upregulation and downregulation of FADD reported in various cancer types [115]. Additional in vitro studies have similarly observed increased FADD expression in OSCC, with FADD upregulation in a Taiwanese cohort being associated with increased risks of lymph node metastasis and poorer overall prognosis [116,117].

### 4.2. Bcl-2 Family Proteins

The Bcl-2 family is a large group of proteins consisting of pro-apoptotic proteins BAX, BAK, and the BH3-only subfamily, along with anti-apoptotic members such as Bcl-2, Bcl-xL, and Bcl-W, many of which are implicated in cancer development [106]. As an anti-apoptotic factor, Bcl-2 protein expression in oral cancers is well documented, although large variability exists with inconsistent historical reports of patient sample Bcl-2 expression in less than 10% of oral tumours [118,119] to greater than 50% [120,121,122]. More recent immunohistochemical experiments estimate this value to be closer to 10–30% [98,123]. Unlike Bcl-2, the pro-apoptotic protein, Bax, is reproducibly upregulated in 43–82% of oral tumours with the strongest expression of Bax observed in well-differentiated tumours [118,120,122,124]. Similarly to FADD, the selection of this pro-apoptotic factor may centre around it non-canonical functions; in vitro experiments have demonstrated nuclear localisation of Bax to the promoter region of *CDKN1a* (encoding for cyclin dependent kinase inhibitor 1) accompanied with increased cell proliferation, myofibroblastic differentiation, and migration in primary human lung fibroblasts [125].

### 4.3. Apoptotic Caspase Proteins

The caspases are an essential family of apoptotic cysteine proteases that cleave cellular substrates and drive cell death. Similarly to the Bcl-2 family of proteins, discrepancies exist regarding the expression changes of the caspase family members in oral cancer. Whilst anti-apoptotic inactivating genetic mutations in members such as caspase 8 have been reported in vitro [67,126,127], other studies conversely demonstrate pro-apoptotic upregulation of caspase 8 [128,129], in addition to upregulation of caspases 3 and 9 [130]. Possibly reconciling these inconsistencies, cluster analysis of apoptotic protein expression profiles from 229 OSCC patient-derived tissue samples identified two distinct populations, one of which had increased expressions of pro-apoptotic proteins and one which was conversely anti-apoptotic [131]. While these two distinct clusters were identified, the functional significances of these pro-apoptotic and anti-apoptotic states in cancer development remain unclear as no significant differences in clinical pathology or disease-free survival could be attributed between them. The existence of a pro-apoptotic state in oral cancer appears contradictory to classical cancer hallmarks, and differentiating the roles of these two states poses an area of potential future investigation. Similarly to FADD and Bax as discussed previously, it may be that these pro-apoptotic states confer some survival advantage through non-classical roles, with emerging evidence suggesting that caspases have many additional roles outside apoptosis, including migration, differentiation, and even proliferation [132].

It is clear that cellular apoptosis is also aberrantly dysregulated at multiple stages in OSCC, ranging from the receptor level of the extrinsic pathway to the pro-apoptotic and anti-apoptotic factors of the intrinsic pathway, and also at the caspase level where all three primary pathways ultimately converge. Questions still exist, such as the biological rationale behind subsets of OSCC that present a pro-apoptotic profile, and additional research is required to characterize the pro-apoptotic stage at a molecular level in order to improve treatment and clinical outcomes. We also note that there are hardly any in vivo studies addressing the molecular mechanisms of OSCC. Several of the early in vivo models did not adequately represent human disease [133] and were focused on pharmacological treatments. A very recent study from the group of Rodini [134] using cancer stem cell subpopulations in mouse models appears to show promise for studying the pro-apoptotic state.

## 5. Interplay between Apoptotic and Proliferative Signalling Networks

The signalling events of the apoptotic and proliferation machinery do not occur in isolation. A high degree of signalling overlap between these two networks exists such that mutations in one will invariably affect the other. The stimulation of ERK has an anti-apoptotic effect by phosphorylation and subsequent proteasome-mediated degradation of the pro-apoptotic BH3-only protein Bim [106]. In the PI3K-AKT-mTOR axis, AKT has analogous anti-apoptotic functions by phosphorylating and inactivating both Bad and caspase 9 [135]. As both ERK and AKT are downstream effectors of Ras which, as previously discussed, is commonly mutated and/or amplified in oral cancer, the overlap and potential for apoptotic resistance through proliferation network dysregulation rapidly becomes apparent. Chief among these is perturbation to the tumour suppressor p53, which is able to induce the expression of the pro-apoptotic BH3-only proteins Puma and Noxa [89] and inhibit the anti-apoptotic effects of Bcl-xL and Bcl-2 [136]. In this regard, mutations to p53 in cancer not only result in enhanced cell proliferation by bypassing cell cycle checkpoints but also confer an additional degree of apoptotic resistance. This is most clearly captured in the use of 4-nitroquinoline 1-oxide (4NQO), a carcinogen that promotes intracellular oxidative stress and genomic instability, in chemically induced animal models of OSCC. The mechanism of action of 4NQO has been reviewed elsewhere [137]. An early 4NQO rat model found *p53* mutations in 55% of early cancer lesions, with a corresponding significant increase in anti-apoptotic Bcl-2 expression as measured by immunohistochemistry [138]. Additional 4NQO induced OSCC mouse models have demonstrated increased EGFR expression, reduced p16 expression, and frequent mutation to *CASP8*, encoding for apoptotic caspase 8 [139,140]. A clear enrichment of apoptotic and proliferative dysfunction in these animal models highlights the salience of these overlapping networks in the development of OSCC.

## 6. Method of Data Collection

Publicly available search engines (PubMed and Google Scholar) were used [141,142], with the following key words: oral cancer, proliferation, proliferation factors, apoptosis, apoptotic factors, mutations, and tumour suppressor. These were additionally searched in combination with specific gene/protein names when reviewing the literature of specific pathways (e.g., PI3K-AKT-mTOR pathway). Articles with citations > 2 and journal impact factor > 1 were considered and then assessed for relevancy and “scientific merit” based on the evaluation of abstract text [143], followed by article content. The parameters were selected to capture as broad a preliminary literature dataset as possible, as oral cancer is not well studied and several studies are recent, i.e., in the last 5 years. Those that pertained to the molecular biology aspects of the proliferative/apoptotic pathways specifically in oral cancer (as opposed to head and neck cancer) were considered for this review. As there were few recent in vivo studies, selected older references were included subsequently.

## 7. Conclusions and Challenges for Future Research

The molecular biology of oral cancer is becoming clearer through the ongoing accumulation of research in the area, although particular aspects still remain elusive. Specifically, interacting pathways in oral tumors of different origin would be of great interest to oral cancer researchers. Here, we have aggregated findings from recently published literature to serve as a quick point of reference for the perturbations in canonical proliferation and apoptotic pathways in OSCC. Where individual studies focus on specific or more focused components of these pathways, this review adds value by presenting a synthesis of this information in the context of entire canonical networks, allowing readers to readily identify how specific perturbations in OSCC exert influence over other proteins in these pathways.

While much of the current review has explored proliferation/apoptosis at the genetic and protein levels, epigenetic alterations including DNA methylation, histone modification, and the role of miRNA have also been studied in oral cancer, albeit to a lesser degree. The current state of epigenetic research in oral cancers has been reviewed elsewhere [144]. In addition, the role of commensals and inflammatory proteins on cancer cell proliferation and apoptosis also serve as a possible direction for future research. Indeed, we have recently observed that the presence of bacterial antigens interact and act as potential confounders in oral cancer proliferation [145,146], affecting both proliferation and apoptotic pathways [147]. Whilst questions clearly still exist, we have presented here a concise review of the primary canonical proliferation and apoptosis pathways and the ways in which they are affected during OSCC development. Further research of these proliferation/apoptotic proteins and pathways will prove invaluable in finding novel markers for prognosis and early diagnosis and in identifying potential targets for novel pharmacological agents that will help restrict the progression of oral cancer and hopefully improve patient outcomes.

## Figures and Tables

**Figure 1 ijms-23-01562-f001:**
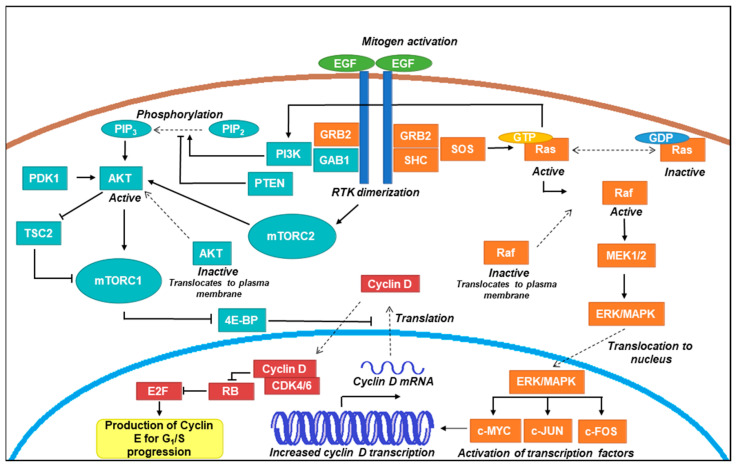
Schematic representation of common proliferative signalling pathways. An overview of proliferative signal transduction through classical Ras-Raf-MEK-ERK/MAPK and PI3K-AKT-mTOR pathways is presented here. Receptor tyrosine kinases (RTKs) such as epidermal growth factor (EGRF) dimerise upon ligand activation (such as through EGF) and recruit proteins containing Src homology (SH) 2 domains such as GRB2 and SHC [32]. In the Raf axis, son of sevenless (SOS) activates Ras through guanine triphosphate (GTP) exchange [33], which subsequently recruits Raf to the plasma membrane where it becomes activated [34]. Raf initiates a signalling cascade by phosphorylating MEK1/2, which in turn phosphorylates ERK (also known as mitogen activated kinase; MAPK) [35]. ERK/MAPK translocates to the nucleus where it activates transcription factors (e.g., c-MYC, c-JUN, and c-FOS), which increase the transcription of cyclin D mRNA [36]. In the AKT axis, RTK dimerization activates PI3K, which stimulates the production of phosphatidylinositol-3,4,5-triphosphate (PIP3). PIP3 production recruits AKT to the plasma membrane where phosphorylation events by PDK1 and mechanistic target of rapamycin complex (mTORC) 2 result in its activation [37,38]. AKT is able to activate mTORC1 and inhibit the action of TSC2, which is a negative regulator of mTORC 1. mTORC1 inhibits 4E-BP, which is a negative regulator of translation [39]. These events result in the production of cyclin D, which complexes with cyclin dependent kinases (CDKs) 4 and 6 to inhibit retinoblastoma (RB) protein and allow E2F transcription factors to produce cyclin E [40,41]. Cyclin E production drives the cell through the G_1_/S transition of the cell cycle towards G_2_ and mitosis [42].

**Figure 2 ijms-23-01562-f002:**
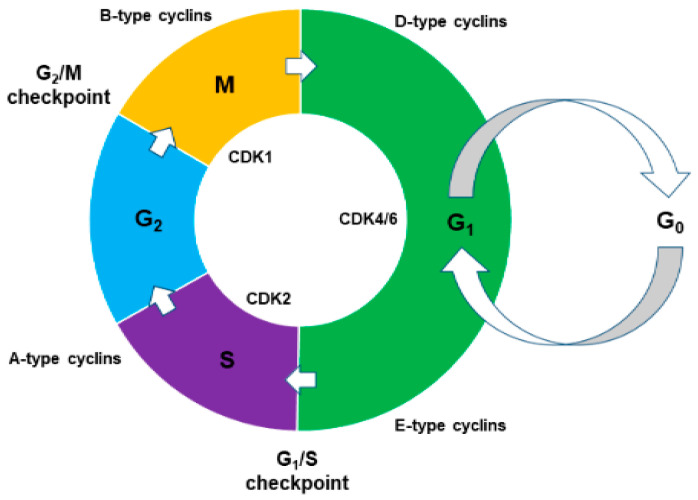
Schematic representation of the cell cycle. Reversibly quiescent cells (G_0_) are able to re-enter the cell cycle and begin cycling in Gap 1 (G_1_) phase upon receiving proper mitogenic signalling. This causes the upregulation of D-type cyclin production which complex with cyclin dependent kinases (CDKs) 4 and 6 and partially inactivates retinoblastoma protein (RB) [42]. This allows for the production of E-type cyclins that interact with CDK2 to hyper-phosphorylate and fully inactivate RB. This inactivation results in the expression of E2F family transcription factors which upregulate the expression of DNA replication proteins, such as DNA polymerase in preparation for Synthesis (S) phase [40]. In the event of DNA damage, sensor kinases ataxia-telangiectasia mutated (ATM) and ataxia-telangiectasia and Rad-3 related (ATR) phosphorylate p53 and the checkpoint kinases Chk1 and Chk2 which collectively signal effectors such as cyclin-dependent kinase inhibitors (CKIs) that arrest the cell at G_1_/S to allow for DNA repair or, otherwise, initiation of apoptosis [83]. A-type cyclins are produced in late S phase and drive mitosis onset, followed by their degradation and production of B-type cyclins that complex with CDK1 and predominantly drive the cell through Mitosis (M) phase [42].

**Figure 3 ijms-23-01562-f003:**
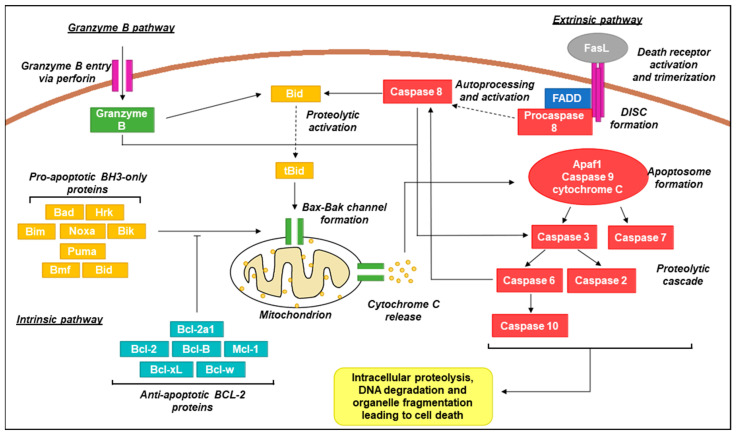
Schematic representation of canonical apoptotic pathways. In the intrinsic pathway, pro-apoptotic BH3-only proteins act as sensors and interact with anti-apoptotic Bcl-2 proteins upon activation by stress signals. Activation over a critical threshold overcomes the anti-apoptotic effects of Bcl-2 proteins and promotes oligomerization of Bax/Bak channels in the mitochondrial membrane that permit the release of the intermembrane space protein cytochrome C [104]. Cytoplasmic cytochrome C promotes apoptosome formation (complex of Apaf1, caspase 9, and cytochrome C) which activates caspases 3 and 7. This results in a signalling cascade providing activation of additional caspase family members which proceed to act on a wide range of cellular targets, ultimately resulting in cell death [105,106]. In the extrinsic pathway, activation of death receptors (e.g., Fas, tumour necrosis factor receptor 1; TNFR1) by cognate ligands (e.g., Fas ligand;FasL, tumour necrosis factor alpha; TNF-α) recruits Fas-associated protein with death domain (FADD) adaptor proteins and procaspase 8 to form the death-inducing signalling complex (DISC) [107]. Procaspase 8 molecules aggregate resulting in autoprocessing and subsequent activation. Active caspase 8 activates caspase 3 and Bid (in its active truncated form; tBid) and converges with the intrinsic pathway via mitochondrial Bax-Bak channel formation [108]. In the granzyme B pathway, granules containing granzyme B and perforin are released from immune cells such as cytotoxic T lymphocytes and natural killer (NK) cells. Perforin oligomerises in the target cell membrane, allowing for entry of granzyme B which is also capable then of activating caspase 3 and Bid, similar to caspase 8 [109].

**Table 1 ijms-23-01562-t001:** Summary of proliferation proteins and their role in oral cancer.

Protein	Role in Oral Cancer	Reference
EGF	Modulates growth and differentiation of oral cancer cells	Bernades et al. [43]
ErbB proteins (EGFR, ErbB2, ErbB3, ErbB4)	Progression and pathogenesis of OSCC	Bernades et al. [43]
GRB2	Overexpression is correlated with lymph node metastasis	Li et al. [44]
Shc	Activation of Shc results in couple β6 signaling to the Raf-ERK/MAPK pathway	Li et al. [45]
SOS1	Useful biomarker in OSCC	Baltanas et al. [46]
Ras	Ras mutations confer therapeutic resistance	Batta et al. [47]
Raf	Oral cancer behaves similarly like other cancer in terms of perturbations in Raf kinase inhibitor protein	Hallums et al. [48]
MEK/ERK	Related to chemoresistance in oral cancer	Kashyap et al. [49]
MAPK	Promotes tumour cell proliferation and anti-apoptosis	Peng et al. [50]
PI3K/Akt/mTOR	Overexpression related to poor prognosis of oral cancer	Harsha et al. [51]
PTEN	Epigenetic biomarker in OSCC	Sushma et al. [52]
PDK1	Targeting PDK1 sensitizes NOTCH1 to PI3K/mTOR pathway	Sambandam et al. [53]
4E-BP	Reactivated by mTOR inhibition in OSCC	Wang et al. [54]
Cyclin D	Early biomarker in oral cancer	Ramakrishna et al. [55]
CDK4, CDK6	The index scores of CDK4, CDK6, and cyclin D1 are associated with the transformation from pre-cancer to oral cancer stage	Kujan et al. [56]
Cyclin E	Related to progression of cancer	Moharil et al. [57]
Cyclin A, Cyclin B	Prognostic significance in OSCC	Monteiro et al. [58]
CDK1, CDK2	CDK1 serve as a prognostic marker and malignant degree for the survival of OSCC patients. CDK2 overexpression serve as a prognostic marker of OSCC	Chen et al. [59]; Mihara et al. [60]
Pocket proteins (RB, RBL1/p107, RBL2/p130)	Suppress OSCC	Shin et al. [61]
ATM and ATR	Related to radiosensitivity in OSCC	Parikh et al. [62]
p53	Inactivated in OSCC resulting in prolonged cell cycle	Ragos et al. [63]
CIP/KIP family (p21, p27, p57)	Cell cycle regulators	Perez-Sayans et al. [64]
INK4 family (p15, p16, p18, p19)	Related to transformation from precancer to oral cancer stage	Agarwal et al. [65]

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
