# Peer review of "Proliferation and Apoptosis Pathways and Factors in Oral Squamous Cell Carcinoma"

_ijms, 2022, doi:10.3390/ijms23031562_

Round 1

Reviewer 1 Report

The paper is well written and provides a valuable lesson. The topic is important and timely. In my opinion standard treatment as well as multidisciplinary management of toxicities and follow-up strategies should be described. Therefore, introduction would be enhanced by addition of several references, such as PMID: 33239190, PMID: 25479896 and PMID: 28427504 to better contextualize the issue at hand in oncologic scenario. The “method of data collection” section needs to be expanded. The quality evaluation of the included studies should be considered.

Author Response

The authors would like to thank the anonymous reviewers for their feedback and valuable comments, ultimately allowing for a publication of higher quality. We have gone through each individual comment and now revised the manuscript to address their concerns, wherever possible with 8 additional references. The rationale behind each change has been included here, where additional information has been provided for each specific comment made the two reviewers. In the proposed publication, we have highlighted the revisions in different colours to allow for ease in tracking the changes made.

Changes based on Reviewer 1’s comments have been highlighted in blue.

Changes based on Reviewer 2’s comments have been highlighted in red.

Reviewer 1: (Changes highlighted in blue font)
Comment 1
In my opinion standard treatment as well as multidisciplinary management of toxicities and
follow-up strategies should be described. Th erefore, introduction would be enhanced by
addition of several references, such as PMID: 33239190, PMID: 25479896 and PMID:
28427504 to better contextualize the issue at hand in oncologic scenario.
Response
Thank you for this comment and for providing additional resources. We agree that some
mention of treatment options should be included in the introduction to provide basic context.
As the focus of this publication is the proliferation and apoptotic signalling pathways affected
in OSCC, we feel going into further detail in to treatments is beyond the scope of this paper,
particularly as the publication has intentionally not included an exploration of the therapeutical
chemicals/agents available and their impacts on the proliferation/apoptotic signalling networks.
The following text has been added to the introduction with three new references, as suggested.
As mentioned in the comment, this is to help provide some basic context into the treatment
options available for OSCC.
Lines 34-38: “Various treatment options exist, including surgery, radiotherapy, chemotherapy,
or a combination thereof, often accompanied with post-operative follow-up and monitoring
[6,7]. Treatment of OSCC during its early stagesshows the most favourable prognosis, with an
estimated 92% 3-year survival rate [8], highlighting the vital importance of early detection.”
Comment 2
The “method of data collection” section needs to be expanded. The quality evaluation of the
included studies should be considered.
Response
The “method of data collection” section has been further expanded as based on feedback.
Additional information regarding the collection of data/literature has been included , along with
three references supporting our method of literature review, which also addresses the scientitif
quality of the studies selected:
Lines 364-375: “Publicly available search engines (PubMed and Google Scholar) were used [132,133], 
with  the  following  key  words:  oral  cancer,  proliferation,  proliferation  factors,  apoptosis,  apoptotic 
factors,  mutations,  and  tumour  suppressor.  These  were  additionally  searched  in  combination  with 
specific gene/protein names when reviewing the literature of specific pathways (eg. PI3K‐AKT‐mTOR 
pathway). Articles with citations > 2 and journal impact factor > 1 were considered, and then assessed 
for relevancy and “scientific merit” based on evaluation of abstract text [134], followed by the article 
content. These parameters were selected to capture as broad a preliminary literature dataset as possible, 
as oral cancer is not extensively studied and several studies are recent i.e. in the last 5 years, with few 
citations till date. Those that pertained to the molecular biology aspects of the proliferative/apoptotic 
pathways  specifically  in  oral  cancer  (as  opposed  to  head  and  neck  cancer)  were  considered  for  this 
review.”

Reviewer 2 Report

The review paper titled "Proliferation and Apoptosis pathways and factors in Oral Squamous cell carcinoma" describes the role of proliferation and apoptosis pathways and factors associated with the progression of oral squamous cell carcinoma (OSCC). Overall, the idea is exciting, but much work needs to be done on the review. Sometimes the review looks like reading a book chapter and digresses from the main objective. A lot of work needs to be done on the review and some issues need to be resolved before it is accepted for publication.

Some comments follow:

- Please reword the abstract: especially lines 18-20. In this review, the authors do not address pharmacological agents that can inhibit proliferation/apoptosis of OSCC cells.

- The authors describe the proliferation and apoptosis pathways involved in OSCC in a very concise manner. In my opinion, the review paper should also include a perspective and synthesis of the new story from the existing data. I mean, the authors limit themselves to listing "facts" of the relevant articles without analysing them and drawing conclusions (lack of critique). You should also end each paragraph with a two to three-line conclusion.

- The authors have mainly discussed the in vitro data, but they should also discuss the consistency of the changes in cell parameters with in vivo tumour progression, along with some reports on whether these results have also been validated in cancer patients.

- The authors should discuss about mutations in different genes involved in apoptosis/proliferation pathways or interacting pathways in oral tumours of different origins, not only OSCC and some therapeutics, if anyone reported in the literature.

The authors should investigate different epigenetic factors in dysregulation of apoptosis/proliferation pathways in oral cancer, including patient data.

- The authors should analyse in detail the role of various non-coding RNAs such as miRNAs in the dysregulation of apoptosis/proliferation-activated/inhibited signalling pathways and the resulting altered pathophysiology of oral cavity cancer.

The authors should reveal changes in the methylation pattern in apoptosis/proliferation pathway genes in oral cancer cells or tumour tissue from mice or patients.

- Please remove vertical lines and unnecessary horizontal lines from Table 1.

- Include tables and figures summarising the main findings. The figures should preferably show the mode of action or general schemes of pharmacological activity.

- Try to clearly separate clinical results from experimental studies (i.e. studies on cells, in vitro, animals) in your review article.

- There are some grammatical problems throughout the manuscript and some statements could be worded more concisely.

Author Response

The authors would like to thank the anonymous reviewers for their feedback and valuable comments, ultimately allowing for a publication of higher quality. We have gone through each individual comment and now revised the manuscript to address their concerns, wherever possible with 8 additional references. The rationale behind each change has been included here, where additional information has been provided for each specific comment made the two reviewers. In the proposed publication, we have highlighted the revisions in different colours to allow for ease in tracking the changes made.

Changes based on Reviewer 1’s comments have been highlighted in blue.

Changes based on Reviewer 2’s comments have been highlighted in red.

Reviewer 2: (Changes highlighted in red font)
Comment 1
Please reword the abstract: especially lines 18-20. In this review, the authors do not
address pharmacological agents that can inhibit proliferation/apoptosis of OSCC cells.
Response
We agree that our review does not address pharmacological agents.
The following text, from lines 18-20, has been removed “for the development of new
prognostic markers and pharmacological agents to combat oral disease” to clarify that the
review will not be discussing pharmacological agents and their roles in OSCC
proliferation/apoptosis.
The revised now text reads:
Lines 19-21: “We anticipate the use of information provided through this review will further progress
translational cancer research work in the field of oral cancer.”
Comment 2
The authors describe the proliferation and a poptosis pathways involved in OSCC in a very
concise manner. In my opinion, the review pa per should also include a perspective and
synthesis of the new story from the existing data. I mean, the authors limit themselves to listing
"facts" of the relevant articl es without analysing them and drawing conclusions (lack of
critique).
You should also end each paragraph with a two to three-line conclusion.
Response
Thank you for the kind words. We set out to write a review that was concise and able to be
used as a quick reference for information pert aining to specific axes/levels of the primary
proliferation and apoptosis pathways.
Some synthesis/perspective has been provided in our original submission.
e.g. Lines 281-284 “This effect is two-fold; OS CC cells are able to avoid cell death not only
by down-regulating expression Fas, but also by upregulating and secreting FasL positive
vesicles to induce apoptosis of T lymphocytes, which normally act as anti-cancer agents.”
Lines 334-342 “While these two distinct clusters were identified, the functional significance of
these pro- and anti-apoptotic states in cancer development remain unclear as no significant
differences in clinical pathology or disease-free survival could be attributed to them.
The existence of a pro-apoptotic st ate in oral cancer appears diffi cult to fit into the classical
cancer hallmarks, and differentiating the roles of these two states poses an area of potential
future investigation. Similar to FADD and Bax as discussed previously, it may be that these
pro-apoptotic states confer some survival advantage through non-classical roles, with emerging
evidence suggesting that caspases have many additional roles outside apoptosis, including
migration, differentiation and even proliferation [127]”) we have avoided too much speculation
in order to present a review that is able to be used for quick reference ofexperimentally verified
findings.
Revisions to address this comment are as follows:
Lines 282-285: “The resultant effect is two-fold, with OSCC cells being able to avoid cell death
not only by down-regulating Fas expression, bu t also by upregulating and secreting FasL
positive vesicles to induce apoptosis of T lymphocyt es, which normally act as anti-cancer
agents.”
Line 300: “Additional in vitro studies have....”
Regarding the last component of this comment, new short conclusions have been added to the
text at the end of the proliferation and apopt osis sections to summarise the information
presented.
Lines 235-240: “While the complex cellular proliferation network is tightly regulated, it is clear
in OSCC that mutations can occur at many junctions within this process. This includes at the
level of mitogenic signalling, during downstream signal transduction through multiple pathway
axes or at the cell cycle level predominantly by altering the tumour suppressor activity. A
greater understanding of the most commo n mutations will undoubt edly benefit the
development of new OSCC treatment options.”
Lines 344-347: “ It is clear that cellular apoptosis is also aberrantly dysregulated at multiple
stages in OSCC, ranging from the receptor level of the extrinsic pathway, the pro-apoptotic
and an-ti-apoptotic factors of the intrinsic pathway, and also at the caspase level where all three
primary pathways ultimately converge. Questions still exist, such as th e biological ra-tionale
behind subsets of OSCC that present a pro-apoptotic profile, and additional re-search is
required to characterize the pro-apoptotic stage at a molecular level, to improve treatment and
clinical outcomes.”
Comment 3
The authors have mainly discussed the in vi tro data, but they should also discuss the
consistency of the changes in cell parameters with in vivo tumour progression, along with some
reports on whether these results have also been validated in cancer patients.
Response
We have added additional text and two references to address this:
Lines 347-351: “We also note that there are hardly anyin vivo studies addressing the molecular
mechanisms of OSCC. Several of the early in vivo models did not adequately represent human
disease [128] and were focused on pharmacological treatments. A very recent study from the
group of Rodini [129] using cancer stem cell subpopulations in mouse models appears to show
promise for studying the pro-apoptotic state.”
Comment 4
The authors should discuss about mutations in different genes involved in
apoptosis/proliferation pathways
or interacting pathways in or al tumours of different orig ins, not only OSCC and some
therapeutics, if anyone reported in the literature.
Response
We have already discussed the mutations in different genes involved in apoptosis/proliferation
pathways. As an example, in lines 161-174 “Being a key regulator, Ras is genetically
deregulated in over 20% of oral cancers, either through genetic mutation and/or amplification
[68]. By contrast, Raf and MEK, both downstream effectors of Ras, sh ow drastically lower
mutability in oral cancer. Through exon sequencing of a small cohort of OSCC samples, a
mutation rate of 2.4% has been re ported for Raf [69], with little or no data reported on MEK
mutation in oral cancer to date. In contrast to Raf and MEK, extracellular signal-regulated
kinase (ERK, also known as m itogen-activated protein kinase ; MAPK) has been reported to
have over 100 downstream cytoplasmic and nucleartargets, including transcription factors that
drive the expression of D-type cyclins which initiate the cel l cycle [33,70]. Whilst
overexpression of ERK has been reported in OSCC [5,71], the reported incidence of this is low
compared to mutations in the RAS gene [72]. In terestingly, in silico pathway analysis found
that both ERK/MAPK and RA S (along with AKT and mTOR) expression levels were
substantially lower in pre-cancerous OLK compared to OSCC, leading to speculation that
mutation in these particular genes results in acquisition of the cancer phenotype [5].”
Regarding interacting pathways in oral tumours of different origin, this is beyond the scope of
the current review, which primarily aims to explore the proliferation and apoptosis pathways
in OSCC, as it is the most common form of oral cancer.
We have added this aspect as a potential further area of research in the concluding paragraph:
Lines 381-384: “The molecular biology of oral canceris becoming clearer through the ongoing
accumulation of research in the area, althou gh particular aspects still remain elusive.
Specifically, interacting pathways in oral tumours of different origin would be of great interest
to oral cancer researchers.”
Comment 5
The authors should investigate different epigenetic factors in dysregulation of
apoptosis/proliferation pathways in oral cancer, including patient data.
Response
Our original intention was to present background information on the canonical proliferation
and apoptosis pathways, and then tie these to recent research that has been conducted
specifically pertaining to oral cancer in a concise manner. In service of this, we omitted the
epigenetic perspective in favour of a review that was more focussed in its scope. Although
passing mention of DNA methylation was mentioned in Line 229-231, “transcriptional
silencing due to high frequency hypermethylation in promoter regions strongly implicating
CDKN2A in oral carcinogenesis [83]”, we wanted to largely avoid re-treading this area in a
diluted form as an excellent review on the current insights in to oral cancer epigenetics had
already been recently published in IJMS in 2018 (“Current Insights into Oral Cancer
Epigenetics”;PMID: 29495520).
However, we appreciate that there is still value in more prominently mentioning these concepts
in our proposed publication for broader contextualisation in the oral cancer research space; the
current text amendment addresses this with th e audience directed to further reading (the
previous 2018 review article mentioned) if interested.
Regarding epigenetics (including DNA methylation and miRNA), the following addition to the
text has been made:
Lines 384-388: “While much of the current review has e xplored proliferation/apoptosis at the
genetic and protein level, ep igenetic alterations including DNA methylation, histone
modification and the role of miRNA have also been studied in or al cancer, albeit to a lesser
degree. The current state of epigenetic research in oral cancers has been reviewed elsewhere
[135].”
Comment 6
The authors should analyse in detail the role of various non-coding RNAs such as miRNAs in
the dysregulation of apoptosis/proliferation-activated/inhibited signalling pathways and the
resulting altered pathophysiology of oral cavity cancer.
Response
Please see the response to Comment 5 concerning epigenetics
Comment 7
The authors should reveal changes in the methylation pattern in ap optosis/proliferation
pathway genes in oral cancer cells or tumour tissue from mice or patients.
Response
Please see the response to Comment 5 concerning epigenetics
Comment 8
Please remove vertical lines and unnecessary horizontal lines from Table 1.
Response
These rectifications have been done.
Comment 9
Include tables and figures summarising the main findings. The figures should preferably show
the mode of action or general schemes of pharmacological activity.
Response
Two tables showing the effect of proliferation and apoptotic proteins in oral cancer have been
included in the proposed publication. We wish to reiterate that the ro le of pharmacological
agents here is beyond the scope of the current submitted manuscript.
Comment 10
Try to clearly separate clinical results from experimental studies (i.e. studies on cells, in vitro,
animals) in your review article.
Response
As this review does not explore pharmacological/therapeutical agents and their effects, no
clinical studies were examined. The majority of the literature cited here was performed in vitro
using tissue samples from oral cancer patients, unless stated otherwise in the text.
To add some clarity however to the nature of some of these experiments, the following
additions have been made to clearly denote in vitro experimental studies.
Line 157: “Increased transcription of TGFA, encoding for the cognate ligand TGF α, has also
been reported in vitro in oral cancers [68]”
Line 183: “In an in vitro next-generation sequencing study, mutations in PIK3CA – encoding
for the p110 catalytic subunit of PI3K – were observed in 7% of 170 or al pre-cancer patient
samples, with additional independent seque ncing studies of 279 oral cancer samples
consistently identifying PIK3CA among the top mutated genes [67]”
Line 224: “The TP53 and RB1 genes, which encode p53 and RB respectively, are commonly
observed in vitro to be structurally altered in patient oral cancer samples [67], with an estimated
29% of pre-cancerous oral lesions demonstrating mutations to TP53 [81]”
Line 280: “Expression of the Fas receptor is s uppressed in OSCC, with two separate studies
reporting detectable Fas in less than 5% of experimental OSCC samples in vitro [105,106]”
Line 301: “Additional in vitro studies have similarly observed increased FADD expression in
OSCC”
Line 329: “Whilst anti-apoptotic inactivating genetic mutations in members such as caspase 8
have been reported in vitro”
References to in vivo studies have been addressed in our response to Comment 3.
Comment 11
There are some grammatical problems throughout the manuscript and some statements could
be worded more concisely.
Response
To the best of our ability, the text has been edited and proof-r ead by native English speakers.
In addition to the edits already mentioned regarding other comments, some re-
wording/grammatical edits have been made to improve clarity and readability of the proposed
publication, most notably to the conclusion of the proposed publication.
Lines 264-267: Apoptosis is the process of program med cell death of which three main
pathways exist; the intrinsic, extrinsic, and granzyme B pathways. All three pathways
ultimately result in the activation of caspase pr oteins that trigger a proteolytic cascade to
dismantle and remove the dying cell [90].
Lines 282-284: The resultant effect is two-fold; OSCC cells are able to avoid cell death not
only by down-regulating Fas expression, but also by upregulating and secreting FasL positive
vesicles
Lines 381-398: “The molecular biology of oral canceris becoming clearer through the ongoing
accumulation of research in the area, although particular aspects still remain elusive. Specifi-
cally, interacting pathways in oral tumours of different origin would be of great interest to oral
cancer researchers. While much of the current review has explored prolifera-tion/apoptosis at
the genetic and protein level, epigenetic al terations including DNA methylation, histone
modification and the role of miRNA have also been studied in or al cancer, albeit to a lesser
degree. The current state of epigenetic research in oral cancers has been reviewed elsewhere
[135]. In addition, the role of commensals and inflammatory proteins on cancer cell
proliferation and apoptosis also serve as a possi ble direction for future research. Indeed, we
have recently observed that the presence of bact erial antigens interact and act as potential
confounders in oral cancer proliferation [136,137], affecting both proliferation and apoptotic
pathways [138]. Whilst questions clearly still exist, we have pr esented here a concise review
of the primary canonical proliferation and apoptosis pathways and the ways in which they are
affected, during OSCC development. Further research of these proliferation/apoptotic proteins
and pathways will prove invaluable in finding novel markers for prognosis and early diagnosis,
and also in identifying potential targets for novel pharmacological agents that will help us
restrict the progression of oral cancer and hopefully improve patient outcomes.”
If there are any additional passages or specific examples of particular concern, we are happy
to make further future edits to the language/wording of the text.

Round 2

Reviewer 1 Report

manuscript has been revised 

Author Response

The authors would like to thank the reviewers again for their time in reading the proposed manuscript. We are happy to see that Reviewer 1 is satisfied with the amendments made during the first round of revisions, and are thankful again for the detailed comments from Reviewer 2 which will help elevate the standard of the proposed publication. Further substantial changes have been made to the manuscript taking on board this new feedback. Our response to Reviewer 2’s comments are highlighted in green, with the addition of nine references, addressing in vivo studies.

Reviewer 1 (Round 2):
Comment 1
manuscript has been revised.
Response
We thank Reviewer 1 for this positive comment. As no specific change s were requested, no
further changes have been made to the second revision of the manuscript.

Reviewer 2 Report

The authors do not answer to my comments/suggestions. Only few answers have been done in a very concise manner. The main concern is represented by the lack of in vivo studies on animals and human tissues. 

This is a major limitation for the readers of the journal. I would suggest the authors to critically check if there are in vivo and human studies conducted to investigate OSCC, otherwise it is quite sufficient to write a review paper discussing only the in vitro data. 

What is the relationship between cell proliferation and apoptosis in OSCC? What does this review add to the data already published in the literature?

Author Response

The authors would like to thank the reviewers again for their time in reading the proposed manuscript. We are happy to see that Reviewer 1 is satisfied with the amendments made during the first round of revisions, and are thankful again for the detailed comments from Reviewer 2 which will help elevate the standard of the proposed publication. Further substantial changes have been made to the manuscript taking on board this new feedback. Our response to Reviewer 2’s comments are highlighted in green, with the addition of nine references, addressing in vivo studies.

Reviewer 2 (Round 2): highlighted in green
Comment 1
The authors do not answer to my comments/suggestions. Only few answers have been done in
a very concise manner. The main concern is represented by the lack of in vivo studies on
animals and human tissues. This is a major limitation for the readers of the journal. I would
suggest the authors to critica lly check if there are in vivo and human studies conducted to
investigate OSCC, otherwise it is quite sufficient to write a review paper discussing only the
in vitro data.
Response
We apologise for misunderstanding Reviewer 2’s comments. We interpreted these comments
to concern implementations of the animal models to study proliferati on/apoptosis – which is
not possible as these networks are heavily perturbed in creation of the models themselves, given
that optimal spontaneous OSCC is exceptionally ra re in laboratory anim als – rather than an
evaluation of the models themselves. While the current review still primarily focuses on recent
in vitro data, additional lines have been added presenting information regarding existing animal
models of OSCC and their proliferation/apoptotic landscapes:
Lines 177-183: “Further compelling evidence for dysregulation of this pathway being involved
in tumorigenesis can be seen in the developm ent of OSCC in transgenic mouse models. Two
models have been described where overexpress ion of KRAS, a member of the Ras family,
coding for the K-Ras protein that is part of the RAS/MAPK pa thway (shown in Figure 1), in
the oral epithelium resulted in the growth of premalignant oral papillomas [75], or dysplasia
and squamous cell carcinoma [76].”
Lines 196-199: “ The importance of PTEN has also be en demonstrated in vivo, where the
inducible loss of PTEN and TGFBR1 (encoding for type I transforming growth factor β
receptor) in trans-genic mice resulted in epith elial hyperproliferation and visible carcinoma
formation 10 weeks after induction with tamoxifen [80].”
Lines 235-241: “The importance of these two tumour suppressor genes has been highlighted in
vivo through transgenic p53-deficien t mice expressing human cyclin D1, an inhibitor of RB,
which develop invasive oral-oesophageal squamous cell carcinoma by approximately 6 months
of age [85]. The formation of well-differentiated squamous cell carcinoma in the oral cavity,
as well as skin, has also been observed more recently in transgenic mice containing only
deletion of p53, although these carcinomas developed later with an average latency of 15-16
months [86].”
Lines 383-393: “This is most clearly captured in the use of 4-nitroquinoline 1-oxide (4NQO),
a carcinogen that promotes intracellular oxidative stress and genomic instability, in chemically-
induced animal models of OSCC. The mechanism of action of 4NQO has been reviewed
elsewhere [135]. An early 4NQO rat model foundp53 mutations in 55% of early cancer lesions,
with a corresponding significant in crease in anti-apoptotic Bcl-2 expression as measured by
immunohistochemistry [136]. Additional 4NQO induced OSCC mouse models have
demonstrated increased EGFR expression, reduced p16 expression and frequent mutation to
CASP8, encoding for apoptotic caspase 8 [137,138]. A clear enrichment of apoptotic and
proliferative dysfunction in these animal models highlights the salience of these overlapping
networks in the development of OSCC.”
We have also added the inclusio n of pertinent in vivo studies in Section 6: Me thod of data
collection as follows:
Lines 405-406: “As there were few recent in vivo studies, select older references were included
subsequently.”
Regarding human tissues , although not strictly in vivo , an overwhelming majority of the in
vitro studies were performed not in cell lines, but by using OSCC tissues from human OSCC
patients. We have already mentioned this in our original manuscript, to which we have now
added the word patient-derived:
Lines 342-345: “Possibly reconciling these inc onsistencies, cluster analysis of apoptotic
protein expression prof iles from 229 OSCC patient-derived tissue samples identified two
distinct populations, one of which had increased expression of pro-apoptotic proteins and one
which was conversely anti-apoptotic [131].”
There are minimal studies performed entirel y in human subjects and centre more on
pharmacological agents, which are beyond the scope of this review.

Round 2

Reviewer 2(round 3)

I have no further concerns about this manuscript.